# STIs during the COVID-19 Pandemic in Hungary: Gonorrhea as a Potential Indicator of Sexual Behavior

**DOI:** 10.3390/ijerph19159627

**Published:** 2022-08-05

**Authors:** Kende Lőrincz, Fanni Adél Meznerics, Antal Jobbágy, Norbert Kiss, Mária Madarász, Laura Belvon, Béla Tóth, Béla Tamási, Norbert Miklós Wikonkál, Márta Marschalkó, András Bánvölgyi

**Affiliations:** Department of Dermatology, Venereology and Dermatooncology, Faculty of Medicine, Semmelweis University, 1085 Budapest, Hungary

**Keywords:** STI, sexually transmitted infection, COVID-19, sexual behavior

## Abstract

The social distancing measures introduced due to the COVID-19 pandemic may have affected the sexual behavior of the population. We collected data retrospectively from the National STD Center of Hungary. The overall patient influx data of the STD Center and the number of patients diagnosed with syphilis, chlamydia, and gonorrhea infections were assessed in the three-month period of 2020 when the strict governmental lockdown was introduced in Hungary. Data were compared to the pre- and post-lockdown quarters of 2020 and matched to the respective quarters of 2018 and 2019. The number of patients diagnosed with syphilis and chlamydia infections in 2020 during the lockdown decreased compared to 2018 and 2019, while the number of gonorrhea cases increased. The lower number of STI screenings resulted in a significant decrease in asymptomatic syphilis and chlamydia case numbers. However, the growing number of gonorrhea cases in 2020 during lockdown highlights that sexual behavior remained unchanged regardless of restrictions. Therefore, gonorrhea may be considered as an indicator of STI incidences during the pandemic.

## 1. Introduction

The first wave of the COVID-19 pandemic imposed immense pressure on the health care system due to the influx of patients diagnosed with COVID-19. Outpatient care was restricted, and non-emergency consultations needed to be postponed [1]. In Hungary, a strict lockdown came into effect on 28 March 2020, which was lifted on 18 June 2020.

The strict social distancing measures imposed due to the COVID-19 pandemic were presumed to have affected the sexual behavior of the population and therefore the incidence of sexually transmitted infections (STIs). A lockdown, when adhered to, leads to fewer instances of casual sexual intercourse; however, it could also motivate more risky sexual behavior by limiting dating options. As a result of decreased access to STI screening, the number of undiagnosed oligosymptomatic and asymptomatic infections could rise, causing an increase in long-term complications, whereas infections with acute symptoms are seen through emergency health care. This phenomenon could lead to different trends in the changes in incidences of various types of STIs during the pandemic. In Greece, Apalla et al. reported a significant decrease in syphilis and gonorrhea case numbers, while in the United States Crane et al. also recorded a drop in the number of syphilis, chlamydia, and gonorrhea cases in 2020 during the lockdown compared to 2019 [2,3]. Whitlock et al. also reported a decrease in the number of gonorrhea cases in the United Kingdom during the lockdown in 2020 compared to 2019 [4].

We aimed to identify the different tendencies in the incidence of syphilis, chlamydia, and gonorrhea during the COVID-19 pandemic’s first wave in Hungary.

## 2. Materials and Methods

### 2.1. Study Design and Data Retrieval

In this single-center retrospective study, data were collected from the hospital information system (HIS) of the National STD Center. We assessed patient influx data and evaluated the number of patients diagnosed with syphilis, chlamydia, and gonorrhea infections separately, from 1 January to 30 September in 2020, and in the same interval of 2018 and 2019. We divided the examined periods into “pre-lockdown” (from 1 January to 27 March), “lockdown” (from 28 March to 18 June) and “post-lockdown” (from 19 June to 30 September) periods in each year, respectively, based on the date of the strict governmental lockdown introduced in Hungary during the first wave of the pandemic.

### 2.2. Laboratory Tests

Syphilis diagnosis was established with nontreponemal (RPR Carbon Kit, Lorne Laboratories Limited, Danehill, UK) and treponemal (SERODIA^®^-TP-PA, Fujirebio Inc. Tokyo, Japan; abia Treponema Ab, AB Diagnostic Systems GmbH, Berlin, Germany) tests. For chlamydia diagnosis, polymerase chain reaction (Chlamydia trachomatis REAL-TIME PCR Detection Kit, DNA-Technology Research & Production, LLC, Moscow, Russia) was carried out from urethral, cervical, anal and pharyngeal specimens, and in the case of lymphogranuloma venereum suspect symptoms, PCR-based genotyping was performed as previously described [5]. Gonorrhea was diagnosed by PCR (Neisseria gonorrhoeae REAL-TIME PCR Detection Kit, DNA-Technology Research & Production, LLC, Moscow, Russia) and cultures from urethral, cervical, anal, and pharyngeal specimens.

### 2.3. Statistical Analysis

The patient influx data of the STD Center and the number of patients diagnosed with syphilis, chlamydia, and gonorrhea separately during the lockdown in 2020 were compared to the pre- and post-lockdown quarterly data of 2020 and the corresponding quarters of 2018 and 2019.

The following ratios were determined: (1) patients diagnosed through STI screening to patients with acute symptoms, (2) latent stage syphilis cases to symptomatic early-stage syphilis (primary and secondary syphilis) and (3) the ratio of L_1_-L_3_ chlamydia serotypes to D-K chlamydia serotypes.

Two-tailed Pearson’s chi-square test was used to compare different quarter-year periods and different patient subgroups, while *p* < 0.05 was considered statistically significant. Statistical analyses were performed using Statistica v13.5.0.17 software (TIBCO Software Inc., Palo Alto, CA, USA).

## 3. Results

### 3.1. Patient Influx

The number of incoming patients of the STD Center decreased to 1154 patients in 2020 during the lockdown from 2490 patients in the same period of 2018 and 2394 patients in 2019. The overall number of patients arriving at the STD Center per month was summarized in Figure 1a. The total number of syphilis cases declined in 2020 during the lockdown period (n = 42) compared to the pre-lockdown period of 2020 (n = 85), and the “lockdown” periods of 2018 (n = 57) and 2019 (n = 71). Chlamydia case numbers also showed a decrease in 2020 during the lockdown (n = 17) compared to pre-lockdown in 2020 (n = 39) and the “lockdown” periods of 2018 (n = 20) and 2019 (n = 30). Surprisingly, an increase in the overall number of gonorrhea cases in 2020 during the lockdown (n = 102) was recorded as compared to the pre-lockdown period of 2020 (n = 99) and the “lockdown” periods of 2018 (n = 61) and 2019 (n = 79), which continued after the lockdown in 2020 (n = 115) (see Figure 1b–d and Table A1 in Appendix A). In the post-lockdown period of 2020, syphilis (n = 114) and chlamydia (n = 31) case numbers increased as well.

### 3.2. Number of Patients Diagnosed by STI Screening

Comparing the ratio of the patients diagnosed with syphilis, chlamydia, and gonorrhea infections by screening to patients with acute symptoms, a significant decrease was recorded during the lockdown in 2020 compared to the period before the lockdown (*p* < 0.0029).

When looked at individually, syphilis and chlamydia cases showed the same tendency. The ratio of patients diagnosed by screening decreased significantly during the lockdown period as compared to the period before the lockdown (*p* = 0.0099 in case of syphilis; *p* = 0.0066 in case of chlamydia) and increased significantly after the lockdown (*p* = 0.022 in case of syphilis; *p* = 0.011 in case of chlamydia). The ratio of gonorrhea cases diagnosed by screening versus acute symptomatic cases did not change significantly (see Table A2).

### 3.3. Syphilis Stages

There was a significantly lower number of asymptomatic latent stage syphilis cases found compared to early-stage syphilis cases during the lockdown in 2020 when data were compared to the period before the lockdown (*p* = 0.017). When data from the lockdown and post-lockdown periods were compared, the increase in case numbers was significant for the latter (*p* = 0.019) (see Table A3).

### 3.4. Chlamydia Serotypes

The proportion of different chlamydia serotypes did not change significantly when the lockdown period of 2020 was compared to the pre- and post-lockdown periods of 2020 and the lockdown periods of previous years. However, when the whole first wave of COVID-19 was examined from 1 January to 30 September, the ratio of L_1_-L_3_ chlamydia serotype to D-K chlamydia serotype decreased significantly in 2020 as compared to 2018 (*p* = 0.0455) and 2019 (*p* = 0.0074) (Table A4).

## 4. Discussion

The lockdowns imposed in the first wave of the COVID-19 pandemic have led to a general decrease in the number of diagnosed STI cases. A significant decrease was reported in the number of syphilis and gonorrhea cases in Greece, the number of gonorrhea cases in the United Kingdom, and the number of syphilis, chlamydia, and gonorrhea cases in the United States [2,3,4].

In our study, patient influx decreased in 2020 during the lockdown, similarly to previous studies [2,3,4]. When syphilis, chlamydia, and gonorrhea case numbers were separately analyzed, a mix of trends was observed: the number of syphilis and chlamydia cases decreased but there was an increase in the overall number of gonorrhea cases during the lockdown, followed by an increase in case numbers for all three diseases.

Syphilis and chlamydia are mainly diagnosed by STI screening due to the usually asymptomatic nature of these diseases, whereas gonorrhea is typically seen with acute symptoms after a short incubation period. The increasing number of gonorrhea cases during the lockdown highlights that the decreased number of STIs could possibly be a pseudo-decrease due to the increase in the number of undiagnosed cases. Consequently, gonorrhea may be considered as an indicator of the incidence of STIs during the pandemic, suggesting that the pandemic had a limited effect on sexual behavior [6].

Due to the lockdown, limited access to STI patient care could have led to fewer STI screenings, which resulted in a decreased ratio of patients diagnosed by STI screening. Whitlock et al. recorded a similar tendency; although the overall number of gonorrhea cases decreased in their practice, the number of symptomatic gonorrhea cases showed no change, which suggests that the decrease only affected the number of asymptomatic cases which otherwise would have been diagnosed by STI screening [4]. The decrease in the proportion of latent syphilis cases, followed by a sharp increase after the lockdown, along with the overall syphilis case numbers, further highlights the importance of vigilance during the pandemic.

The shift towards D-K chlamydia serotypes from L_1_-L_3_ serotypes seen during the nine-month period of 2020 may be the consequence of travel restrictions imposed in Europe in 2020, as lymphogranuloma venereum cases are predominantly imported to Hungary from abroad [5].

Our study is the first to detail the changes in STI incidence during the COVID-19 pandemic in Hungary, providing a unique insight into STI patient care during restrictions. Our main limitation was that the study used a single-center retrospective study design. Additionally, we did not collect data directly related to sexual behavior, but adopted a potential, indirect outcome of sexual behavior and also measured the incidence changes of different STIs during the pandemic.

## 5. Conclusions

The limited number of STI screenings during the lockdown which was introduced due to the first wave of the COVID-19 pandemic resulted in a decrease in syphilis and chlamydia cases. However, the increasing number of gonorrhea cases in 2020 during lockdown highlights that sexual behavior remained unchanged regardless of restrictions. Therefore, gonorrhea may be considered as an indicator of the incidence of STIs during the pandemic. While the fifth wave of the COVID-19 pandemic gradually resolves, it is crucial to summarize the knowledge from the past two years with respect to the sexual behavior of the population. By highlighting the importance of STI screenings, we urge the implementation of protocols that will allow access to STI patient care during a future pandemic.

## Figures and Tables

**Figure 1 ijerph-19-09627-f001:**
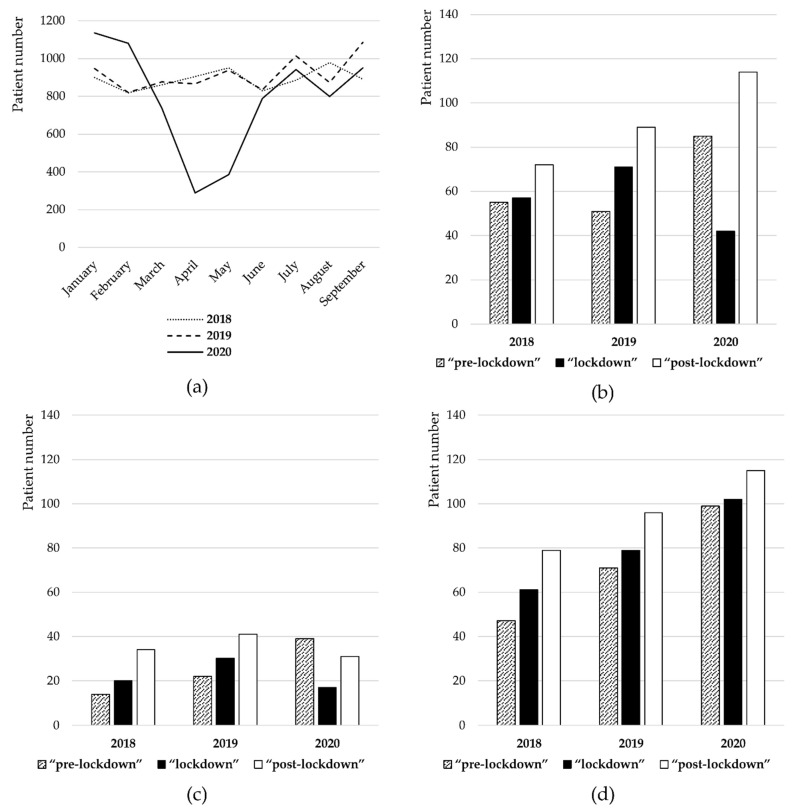
(**a**) The number of STI patients from 1 January to 30 September. (**b**) The number of patients with syphilis in the “pre-lockdown” (1 January–27 March), “lockdown” (28 March–18 June), and “post-lockdown” (19 June–30 September) periods of 2018, 2019, and 2020, respectively. (**c**) The number of patients with chlamydia in the “pre-lockdown” (1 January–27 March), “lockdown” (28 March–18 June), and “post-lockdown” (19 June–30 September) periods of 2018, 2019, and 2020, respectively. (**d**) The number of patients with gonorrhea in the “pre-lockdown” (1 January–27 March), “lockdown” (28 March–18 June), and “post-lockdown” (19 June–30 September) periods of 2018, 2019, and 2020, respectively.

## Data Availability

The data underlying this article are available in the article and in its Appendix A.

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
