# Peer review of "STIs during the COVID-19 Pandemic in Hungary: Gonorrhea as a Potential Indicator of Sexual Behavior"

_ijerph, 2022, doi:10.3390/ijerph19159627_

Round 1

Reviewer 1 Report

This study is very simple but interesting in this pandemic scenario. It certainly only raises hypotheses and the authors clearly reinforce that gonorrhea is symptomatic of STIs, and for this reason it may have remained so frequent in  diagnoses present in the pandemic.

Author Response

We would like to thank you for your precise review and suggestions

Reviewer 2 Report

This is a really interesting paper and certainly an issue that requires attention to understand patterns of STIs in a very unique context. Below I provide some comments for your consideration.

The introduction would benefit from a more detailed review of the literature. While very little research in likely to be available due to the unique context of the COVID 19 pandemic, there needs to be more evidence to support your rationale drawing on theory and other relevant evidence.

Please clarify the denominator. You state daily patient numbers, is this the number of patients who attended the clinic each day for testing / screening or all patients who attended each day for any reason? It is unclear what the cases are a proportion of.

You provide analysis for 1) patients diagnosed by STI screening to patients with acute symptoms, 2) latent stage syphilis cases to symptomatic early-stage syphilis (primary and secondary syphilis) and 3) the ratio of L1-L3 chlamydia serotypes to D-K chlamydia serotypes. However, analysis regarding screening differences across the STIs would be useful also.

Figures b and c appear to show a decline in screening in 2020 as one would expect during lockdowns, however figure d suggests an increase in screening during the same time although you state this is not statistically significant. Provide some commentary about why you think screening for gonorrhoea did not decline like it did for syphilis and chlamydia. Also, provide the data that shows whether the rates of screening versus diagnosed with acute symptoms.

The description of the significant findings in sections 3.2, 3.3 and 3.4 are not provided in data tables or in the figures. These are needed to assist the reader.

Table A1 – part of the issue with chi squares is that it is not clear where the association is, so some more caution is needed in your interpretation on page 3 from line 89. For example, we don’t know the significant association is across years, it could be across lockdown, vs pre or post.

Table A2 and 3  – is it possible to test for the difference to see if any of these changes are significant? Drawing any meaning from raw counts is difficult. It would also be useful if Table A3 included, pre, lockdown and post in line with other tables. You have reported in the text yet there is no data to support this.

It is not apparent from the data presented how the conclusion regarding gonorrhoea was reached. This needs further discussion to demonstrate based on your data. Some caution is also needed about expressing your conclusions as related to sexual behaviour during lockdowns as it is not assessing sexual behaviour but rather a potential outcome of sexual behaviour. There are various factors that should be considered here other than lack of adherence to lockdown requirements.

A statement about the strengths and limitations of the study is required. 

The manuscript would benefit from a detailed proofread and edit to improve readability and grammar.

Reviewer 3 Report

The manuscript entitled ‘STIs during the COVID-19 pandemic in Hungary: Gonorrhea as a potential indicator of sexual behaviour?’ is a short, well written paper, mainly reporting the decrease of chlamydia and syphilis and the unchanged number of cases with gonnorrhea during Covid lock down in 2020 in Budapest, Hungary. This study is based on STI visit data retrieval of a single clinic. It is interesting to see the influence of less STI screening due to lockdown on the most prevalent STI pathogens such as Chlamydia trachomatis (=CT), Neisseria gonorrhoeae (NG) and syphilis. The authors compared data of three episodes before, during and after lockdown in 2020 to the same calendar periods in 2018 and 2019. The message that STI clinics should remain accessible at all times and that screening for STI is important is a view that I share, but this does not necessarily follow from the data in this paper.

A number of shortcomings need to be addressed, to improve the manuscript.

  1. In the introduction is mentioned that the incidence will be studied of the three STIs. My main objective is that only absolute numbers are given and daily number of cases. So incidences (percentages per time period) are certainly not presented. In Figure 1a also the number of patients who visited the STI clinic in all three years during the lock down period is shown. Please use these data to calculate the percentage, or prevalence of positive cases; so for example, number of positive CT cases per number of patient visits. Please add these percentages in Figure 1b, 1c and 1d, and also in the Appendix tables Table A1, Table A2 and Table A3.

  2. In Methods it is not explained how the ‘daily patient numbers’ were calculated. Are these number of STI positive diagnoses per day? This is not very informative. As mentioned above (point 1) it is more informative to calculate the positives per number of patient visits, so percentages. Please explain and adjust.

  3. In Methods it is not explained which laboratory tests were performed to come to the CT, NG and syphilis positive diagnoses. Also L1-L3 (LGV type of CT) serovars the lab test is not explained: were there chlamydia cultures performed and subsequently antibody tests? Or was the ompA gene (partly) sequenced to obtain the CT genovar? Please explain and add the lab tests used.

  4. In Results, line 96,  it is stated that the number of cases with NG remained unchanged in 2020. It is clear however for NG that both the number of symptomatic and the number by screening increased in 2020 relative to 2018 and 2019. Please adjust.

  5. In Results, line 120, is stated that the proportion of LGV cases did not change significantly during the years 2018, 2019 and 2020. In Table A3 it can be seen however that the number of L1-L3 serotypes increased in 2020 relative to previous years. Since LGV is the more invasive infection type of CT one could reason that this infection behaves the same as NG. Please comment.
    Also for LGV it is much more interesting to have percentages, so the number per number of patients who presented at the clinic. Please adjust in Table A3 and text.

  6. In Discussion no limitations of the study are mentioned. It should be noted that this was a single clinic visit (only one STI clinic in Budapest) as limitation. Also other limitations may apply. Please add.

Minor comments

-Line 89: please add ‘s’  after ‘case’

-Line 164: Consider to replace ‘in regards of the sexual’ with ‘with respect to sexual’

-Line 166: Please add ‘care’ after ‘STI patient’ 

Reviewer 4 Report

The manuscript by LÅ‘rincz et al. evaluated tendencies in the incidence of syphilis, chlamydia, and gonorrhea, during the COVID-19 pandemic’s first wave in Hungary. In my opinion, the manuscript contains important information but needs some adjustments before being published.

1. The Introduction needs more basis to justify the importance of the study;

2. What are the limitations of the study?

3. The authors should add to the discussion what could justify the increase in the number of cases of gonorrhea even during the lockdown period.

Round 2

Reviewer 3 Report

Thank you for answering all points raised and adjusting the manuscript where requered. In the Response letter the authors state: "Based on your suggestion we calculated the absolute patient number per patient influx ratios, we attached the table with these results below. However, we think that these values might be biased as the patient flow of the STD Center dropped significantly, and it does not represent well the population at risk – these results show the increase in the ratio of syphilis and chlamydia as well, which we think is a false result. "

I do think that indeed adding the percentages shows clearly another message: also for Syphilis and Chlamydia the rates of positives went up. Tis means indeed as the authors argue, the asymptomatice persons did not show up and those with symptoms did. The persons with synptoms are then clearly those with the highest positivity rates.

I also agree with the authors, however, that it is important to continue screening during lock down, since after the lock down the numbers rose for all STI, indicating that these people were 'hidden' because they could not visit the clinic earlier.